# Porous Carbon–Carbon Composite Materials Obtained by Alkaline Dehydrochlorination of Polyvinyl Chloride

**DOI:** 10.3390/ma15217636

**Published:** 2022-10-30

**Authors:** Yury G. Kryazhev, Irina V. Anikeeva, Mikhail V. Trenikhin, Tatiana I. Gulyaeva, Valeriy P. Melnikov, Vladimir A. Likholobov, Olga B. Belskaya

**Affiliations:** 1Center of New Chemical Technologies BIC, Boreskov Institute of Catalysis, Siberian Branch, Russian Academy of Sciences, 644040 Omsk, Russia; 2Petrochemical Institute, Department Chemistry and Chemical Technology, Omsk State Technical University, 644050 Omsk, Russia; 3N.N. Semenov Federal Research Center for Chemical Physics, Russian Academy of Sciences, 119991 Moscow, Russia; 4Boreskov Institute of Catalysis, Siberian Branch, Russian Academy of Sciences, 630090 Novosibirsk, Russia

**Keywords:** polyvinyl chloride dehydrochlorination, carbon nanocomposites, nanoglobular carbon, graphite oxide

## Abstract

Porous carbon–carbon composite materials (PCCCM) were synthesized by the alkaline dehydrochlorination of polyvinyl chloride solutions in dimethyl sulfoxide containing the modifying additives of a nanostructured component (NC): graphite oxide (GO), reduced graphite oxide (RGO) or nanoglobular carbon (NGC), with subsequent two-step thermal treatment of the obtained polyvinylene–NC composites (carbonization at 400 °C and carbon dioxide activation at 900 °C). The focus of the study was on the analysis and digital processing of transmission electron microscopy images to study local areas of carbon composite materials, as well as to determine the distances between graphene layers. TEM and low-temperature nitrogen adsorption studies revealed that the structure of the synthesized PCCCM can be considered as a porous carbon matrix in which either carbon nanoglobules (in the case of NGC) or carbon particles with the “crumpled sheet” morphology (in the case of GO or RGO used as the modifying additives) are distributed. Depending on the features of the introduced 5–7 wt.% nanostructured component, the fraction of mesopores was shown to vary from 11% to 46%, and S_BET_—from 791 to 1115 m^2^ g^−1^. The synthesis of PCCNC using graphite oxide and reduced graphite oxide as the modifying additives can be considered as a method for synthesizing a porous carbon material with the hierarchical structure containing both the micro- and meso/macropores. Such materials are widely applied and can serve as adsorbents, catalyst supports, elements of power storage systems, etc.

## 1. Introduction

Polyvinyl chloride (PVC) is a polymer product ranking fourth in terms of world output. Its growing production and extending application field (housebuilding, health care, transport, agriculture, information technologies, and textile manufacture) [1,2,3] make the utilization of PVC wastes a pressing problem [3,4,5,6]. Chloropolymer solid wastes cannot be treated by thermal-oxidative methods or pyrolysis like other polymers [3,5,7,8,9,10], because in the temperature range of 500–1200 °C high-toxic substances are formed; in particular, polyaromatic compounds, benzo(a)pyrenes, and dioxins (2,3,7,8-tetrachlorodibenzo-p-dioxin and a large group of structurally and toxicologically related compounds) [1,5,7,11]. Given this connection, it is necessary to develop methods for the removal of covalently bound chlorine from PVC at relatively low temperatures (below 500 °C) [12]. Catalytic dehydrochlorination is one of the promising directions [10]. At the same time, dehydrochlorination of PVC under the action of bases or alkalies remains the main reaction for this purpose [13,14,15,16,17,18]. Thus, hydrothermal treatment of a PVC powder in 1% aqueous solutions of NaOH [13] or Cu(NO_3_)_2_ [14,15] at a temperature up to 240 °C and a corresponding pressure leads to the removal of 94% and 98% chlorine from PVC, respectively; high degrees of chlorine removal from PVC (95–97%) were also achieved upon heating (170 °C and 200 °C) of the mixtures of PVC powders and Zn or Ca oxides [16,17], which, being the bases, served as dehydrochlorinating agents. The indicated approach to the development of a dioxin-free process for utilizing chloropolymeric solid wastes will be economically attractive only if the carbon material (CM) obtained by dehydrochlorination of PVC finds its niche for commercial application; for example, if its structural parameters can hardly be achieved by the existing methods of CM synthesis. Thus, the authors of [18], when synthesizing carbon materials from PVC wastes, performed the dehydrochlorination step in a ball mill in the presence of KOH and thiourea, and the resulting product was thermally treated in a nitrogen flow at 600 °C. The produced carbon material had a high specific surface area (S_BET_ = 1230 m^2^ g^−1^) and a multilevel hierarchical macro/meso/microporous structure.

In our earlier papers [19,20,21,22,23], a similar approach to the synthesis of functional carbon materials from PVC and chlorinated PVC (CPVC) was employed; it was based on dehydrochlorination of the initial polymers in the presence of bases with subsequent thermal treatments of the “carbon-enriched products” (polyvinylenes) formed in the dehydrochlorination step. It was demonstrated that such an approach allowed the synthesizing of original porous carbon materials (PCM) because textural characteristics and composition of functional groups in the produced PCM could be varied by changing the formation conditions of carbon structures (temperature-time modes and composition of the gaseous medium) and also by introducing the modifying additives into the initial chloropolymer due to its thermoplasticity and solubility. We have chosen conditions for the synthesis of microporous carbon materials (S_BET_ = 1000–1200 m^2^ g^−1^) by thermal heating (carbonization) at 400 °C of polyvinylenes obtained by the alkaline dehydrochlorination of CPVC [20] and PVC [21] with subsequent high-temperature (800–1000 °C) steam or carbon dioxide activation (partial burnout of the carbon material under the action of water or carbon dioxide as the oxidants to develop its porosity). As shown in [22,23], pore structure parameters of the PCM obtained from CPVC can be controlled by the introduction of nanodispersed carbon black or reduced graphite oxide as the modifying additives in the reaction medium during dehydrochlorination of the initial polymer.

The current trend is the creation of a wide range of carbon-based nanostructured and composite materials in combination with their theoretical studies, which make it possible to understand the issues of synergy and the processes occurring during synthesis [24,25,26,27,28,29]. The present work aimed to obtain porous carbon–carbon composite materials (PCCCM) using the above-mentioned scheme of carbon materials synthesis from PVC, and to reveal the effect exerted by some nanodispersed carbon additive-components (NCs) introduced during the synthesis on the pore structure of PCCCM. For this purpose, NCs such as nanoglobular carbon (NGC), graphite oxide (GO) and partially reduced graphite oxide (RGO) were introduced into the polymer solution at the PVC dehydrochlorination step. To develop the pore structure in the synthesized PCCCM, a conventional activation method was used: thermal treatment in a CO_2_ medium at ca. 800–1000 °C [20,21], which are the temperatures providing gasification of carbon material and promoting the development of its pore space. Carbon dioxide served also as an inert gas medium during the formation and condensation of polyvinylenes (to prevent the undesirable action of air oxygen on the polyene chain structures), because at relatively low temperatures (up to 400 °C), CO_2_ did not manifest oxidizing ability toward the carbon material. In the structural study of the synthesized PCCCM, particular attention was paid to changes in their pore structure under variation of the NC features. The morphology of the produced PCCCM was examined by transmission electron microscopy (TEM), and parameters of their pore structure by the low-temperature nitrogen adsorption.

## 2. Materials and Methods

The initial polymer material was represented by suspended polyvinyl chloride PVC-S-SI-70 (produced at JSC Sayanskhimplast, Sayansk, Russia) in the form of spherical grains with the size of 100–200 µm and molecular-weight distribution 7000–7100.

Carbon black of P 267-E grade served as a modifying additive with the nanoglobular carbon structure. It was produced by the furnace method (pilot plant production at the Center of New Chemical Technologies BIC SB RAS) and contained 97.8% carbon (hydrogen, oxygen and ash residues as the balance) in the form of 500–800 µm spherical grains.

The modifying additives based on graphite oxides included graphite oxide (GO) with the composition: C—59.2 at.%, O—31.9 at.%, H—8.8 at.%, and partially reduced graphite oxide (RGO) with the composition: C—84.0 at.%, O—10.4 at.%, H—5.6 at.% [30]. We prepared GO using the classical Hummers’ method [30,31] using the oxidation of pencil lead with sodium nitrate and potassium permanganate in concentrated sulfuric acid. GO was purified by multiple washes with a 5% solution of hydrochloric acid, and then with water to remove sulfate and chloride ions. The GO suspension obtained as a swollen paste was dried in a thin layer at 60 °C. RGO was prepared by the “explosive” thermal reduction of GO at 700–750 °C according to the method reported in [23]. The explosive reduction of GO was carried out in a vertical tubular furnace based on a quartz tube with the diameter 50 mm and length 100 cm; an RGO collection container was hermetically attached to its bottom part. The entire volume of the furnace was preliminarily purged with argon. A heating section, 50 cm in length (700–750 °C), was located in the bottom part of the tube. GO plates were thrown into the furnace channel. In a hot zone of the furnace, a large amount of gaseous products was released, which resulted in splitting of the graphite oxide particles, a sharp drop in the oxygen content in the material, and the formation of a graphene-like material, RGO, with the specific surface area 800 m^2^ g^−1^ and higher [23].

An important problem in the synthesis scheme of PCCCM considered above is to ensure close interaction of the modifying additive nanoparticles with the initial polyvinyl chloride molecules (or aggregates of molecules). This problem is of interest, per se; it is considered in many studies where PVC modification by carbon NC is used to render antistatic or shielding properties to NC/PVC composite materials [32,33,34,35,36,37,38,39], to improve thermal or electrophysical properties [39,40,41,42,43,44], and to apply such materials in energy storage devices [41,42], chemical sensors [41,42] and microelectronics [39,44]. A uniform distribution of carbon nanoparticles in PVC melt or its solutions in organic solvents is obtained using ultrasonic treatment [33,40,41,42,44,45], preliminary surface functionalization of the modifying carbon nanoparticles [46,47,48], surfactant additives [49,50], and mechanical mixing in a dry form in special-purpose mills [35,39,42,43] or extruders [33,34]. The analysis of available publications showed that the most efficient contact of the modifying additive with chloropolymer molecules (or aggregates of molecules) during the synthesis of such polymer–carbon nanocomposites is achieved in an ultrasonically treated liquid medium [33,40,41,42,44,45]. In this connection, the following sequence of procedures was accepted for the synthesis of PCCCM: dissolution of PVC in dimethyl sulfoxide (DMSO); introduction of the carbon nanocomponent, which was previously dispersed by ultrasonic treatment in a small volume of DMSO, into the obtained solution of PVC; introduction of the dehydrochlorinating agent (a KOH solution in DMSO) in the produced stable dispersion; and sedimentation, filtering, washing with water and drying of the resulting polyvinylene-NC, followed by its carbonization and activation.

When performing this sequence in our work, the following details should be noted. (i) Dehydrochlorination of PVC was carried out in a 1 wt.% solution of the polymer in DMSO in the presence of KOH (the polymer to alkali ratio of 1:2 wt.) at 20 °C for 6 h under continuous stirring; the obtained product was filtered, washed with water in a Soxhlet apparatus to remove Cl^−^ ions from the filtrate, and dried at 100 °C. (ii) The modifying NC additives, constituting 1% of the PVC weight, were introduced in the polymer solution as the dispersion in DMSO formed by ultrasonic treatment before introducing the alkali; ultrasonic treatment was performed for 60 min using an ultrasonic disperser Bandelin SONOPULS HD 4100; the ultrasonic treatment mode (44 kHz, 100 W, 200 mL) chosen in a series of preliminary experiments allowed us to obtain the dispersions that were stable for 48 h at 20 °C. (iii) Carbonization of the materials was performed in a tubular furnace SNOL 7.2/1100 in two steps: at 200 °C (2 h) and 400 °C (2 h) in a CO_2_ flow. (iv) The resulting carbon materials were activated in the same furnace in flowing CO_2_ (900 °C, 2 h).

Characteristics of specific surface area and pore structure of the tested samples were obtained from the analysis of nitrogen adsorption–desorption isotherms at the liquid nitrogen temperature, which were measured in a vacuum static setup ASAP-2020M Micromeritics. Relative pressures P/P_0_ of the nitrogen vapor ranged from 10^−6^ to 0.996. Before the adsorption measurements, samples under consideration were evacuated at 300 °C for 10 h to a residual pressure not higher than 10^−3^ Pa. Specific surface area (S_BET_) was calculated by the commonly accepted Brunauer–Emmett–Teller (BET) method using the adsorption isotherm in the range of equilibrium relative pressures 0.005 < P/P_0_ < 0.06 [51,52,53]. The pore volume (V_ads_) was estimated from the nitrogen adsorption at P/P_0_ = 0.990 and included both the micro/mesopore volume (d = 2–50 nm) and the macropore fraction (d < 200 nm). The micropore volume (V_micro_) was calculated by a comparative analysis (t-method). The volume of meso(macro)pores was calculated by subtracting the volume of micropores from V_ads_ at a relative pressure of 0.990.

To obtain the differential pore size distribution curves, the standard Barrett–Joyner–Halenda (BJH) method was applied for the adsorption branch, assuming a cylindrical pore model [54], while the non-local density functional theory (NLDFT) method was applied for slit-like pores using the DFT Plus (Micromeritics) software [55].

The morphology of the synthesized materials was examined by TEM using a high-resolution transmission electron microscope JEM 2100 JEOL, Tokyo, Japan (accelerating voltage 200 kV and crystal lattice resolution 0.14 nm) with an energy dispersive X-ray (EDX) spectrometer INKA 250 (Oxford Instruments, High Wycombe, UK). The main source of attention was the analysis of images of local areas of carbon composite materials—mainly the boundaries between components of the composite and adjacent areas. The analysis and electronic processing of TEM images were performed using the Digital Micrograph (Gatan) program and fast Fourier transform (FFT) techniques, which were similar to those employed in [56]. The distances between graphene layers in the synthesized samples were measured using the FFT images of graphene layers and the radial intensity profiles obtained by electronic processing of FFT images. The intensity maximum on such plots corresponded to the inverse interlayer distance. To convert TEM images into contrasting black-and-white ones, i.e., into images without grey shading, the “thresholding image processing” function reported in [57] was applied.

## 3. Results

### 3.1. Synthesis of Carbon–Carbon Nanocomposites

The removal of covalently bound chlorine during the liquid phase or mechanically activated dehydrochlorination of PVC under the action of alkalies (NaOH, KOH) has been studied quite well (see, for example, refs. [13,18,19,20,21,22,23,24]) and can be presented in the general form as a scheme in Figure 1. The synthesis route is described in detail in the experimental section. Thus, the formation of polyvinylenes (Figure 1a,b) is reflected in the Raman spectra as the presence of UV-vis DRS bands at 1503 and 1124 cm^−1^, which belong to the stretching vibrations of conjugated C=C bonds and ordinary C-C bonds, respectively, and the absence of intense signals at 635 and 695 cm^−1^, which correspond to the stretching vibrations of C-Cl bonds in the initial PVC [21]. Carbonization of the formed polyvinylenes, which proceeds in the temperature range of 200–400 °C [20,21,22] via the interchain condensation (cross-linking) of polyvinylene chains (Figure 1b,c), manifests itself as changes in the Raman spectra: the disappearance of the bands typical of polyene chains and the appearance of new bands that are typical of graphitic carbon materials; UV-vis bands at 1356 cm^−1^ (D-band) and 1589 cm^−1^ (G-band) [21] with the intensity ratio I_D_/I_G_ = 0.87, characterizing strongly disordered defect packings of graphene layers [29].

The elemental analysis revealed that polyvinylene produced as a result of alkaline dehydrochlorination of PVC in the DMSO medium contains only 1.8 wt.% chlorine, which means that more than 90% of chlorine was removed from macromolecules of the initial PVC. After a two-step thermal treatment (200 °C, 2 h and 400 °C, 2 h) in the CO_2_ medium, a carbon material with the carbon content of 84 wt.% was formed from polyvinylene, while the content of chlorine dropped to 0.4 wt.% (the rest consisted of hydrogen (4.6 wt.%), oxygen (10.1 wt.%) and sulfur (0.4 wt.%)). A TEM study of this carbon material showed (Figure 2a) that its morphology has a disordered pattern of graphene layers, which is typical of amorphous carbon; see, for example, [57]. The transformation of TEM images into binary images, i.e., into black-and-white images without grey shading (Figure 2b), allowed us to distinguish and emphasize the structural details of graphene layers at the subnanolevel. It can be concluded from Figure 2c that these layers are very short (a length of 1–2 nm), strongly bent in different orientations, and the interlayer distance falls in a range of 0.4–1.0 nm with the distribution maximum at 0.55 nm. These values were calculated from the analysis of radial contrast intensity profiles (RCIP) obtained after processing the FFT images of graphene layers (Figure 2d). Such a structure of the synthesized carbon material particles may be caused by morphological features of the carbon material precursors represented by spherical nanodomains—the aggregates of tangled PVC molecules. Such nanodomains have a diameter of ca. 10–20 nm [59] and are likely to form in a liquid medium due to destruction under the action of DMSO of the globular particles of initial PVC powder, which are the agglomerates of smaller particles. This assumption can also explain the presence of helical structures in Figure 2c, which could be formed by carbon residues of the former tangles of PVC molecules.

As shown by the TEM study of the carbon material obtained after thermal treatment at 900 °C for 2 h (Figure 3), distinct packages of graphene layers were observed in this case. They are significantly extended, which testifies to the formation of a more ordered structure in comparison with the CM after low-temperature carbonization, although the distance between graphene layers virtually does not change and falls in the range of 0.4–1.0 nm with the distribution maximum at 0.65 nm.

In the porous carbon–carbon nanocomposite (PCCNC) synthesis, dehydrochlorination, as well as high-temperature carbonization and activation steps, led to a considerable weight loss of carbon composites (up to 70 wt.%), and the fraction of modifying additives in the final PCCCM increased. Thus, taking into account the weight loss and the assumption that NGC is not gasified under the chosen activation conditions, the estimated NGC content in the composite is ca. 6 wt.%. It should be noted that oxygen-containing GO and RGO (in distinction to NGC, the synthesis of which was performed at high temperatures) undergo changes during the stepwise heating, which are accompanied by the formation of graphitic nanostructured particles and the release of gaseous products. Therefore, the content of such additives in the composites was estimated, taking into account the preliminarily obtained thermal analysis data. According to these data, two thirds of the active oxygen-containing groups decompose in a relatively narrow temperature range of 170–250 °C with equimolar release of CO_2_ and H_2_O, while the remaining third of the active groups decomposes in a wide temperature range of 300–1000 °C with the release of CO_2_ and CO [60]. Taking into account the corresponding weight losses, the content of additives in the obtained composites was 5 and 7 wt.% in the case of GO and RGO, respectively.

### 3.2. A Structural Study of Carbon–Carbon Nanocomposites

A TEM study of the synthesized PCCNC revealed, that for the PCCNC obtained from PVC with the NGC additive, the introduced NC is distributed in the carbon matrix as large aggregates with the average size of ca. 2000 nm, which are represented by splices of primary carbon globules with the diameter of 20–50 nm. Such aggregate splices are typical of the chosen NGC grade (Appendix A). Therewith, the composite synthesis does not lead to the destruction of aggregates or structural transformation of graphene layers in primary carbon globules (the interlayer distance in the incorporated globules is 0.38 ± 0.02 nm (Figure 4c,e) and corresponds to that for the primary globules of the chosen NGC grade). The structure of graphene layers in the carbon matrix also did not undergo substantial changes (Figure 4d), and interlayer distances in the matrix fell in the region of 0.4–1.2 nm with the distribution maximum equal to 0.7 nm (Figure 4f).

A similar pattern was also observed when GO or RGO was used as NC: the obtained PCCNC were also a combination of the carbon matrix and carbon structures of the “crumpled sheet” type distributed in the matrix. Such structures comprise 10–30 graphene layers and generally have a higher structural ordering as compared to the carbon matrix layers (Figure 5 and Figure 6).

Thus, structures of two types were observed in the PCCNC sample with the GO additive: graphitic, i.e., extended graphene layers parallel to each other (Figure 5c), and relatively short and less ordered graphene layers (Figure 5d). Interlayer distances in these structures, which were calculated from the analysis of radial contrast intensity profiles, were equal to 0.35 and 0.65 nm, respectively (Figure 5e,f). Similarly, two types of structures were observed in the CCNC sample with the RGO additive (Figure 6): graphitic, i.e., parallel to each other but short graphene layers (Figure 6c), and less-ordered graphene layers (Figure 6d). In this case, the interlayer distances in such structures were 0.40 and 0.72 nm, respectively (Figure 6e,f).

According to the EDX spectroscopy data, all the synthesized composites are carbon materials with a carbon content not lower than 95 wt.%.

### 3.3. Investigation of the Textural Characteristics of Carbon–Carbon Nanocomposites

Low-temperature nitrogen adsorption–desorption isotherms for the studied samples of carbon materials are shown in Figure 7. Isotherms for the CM synthesized from PVC both in the absence of modifying additives and in the presence of NGC additive are characterized by the steep uptakes at very low P/P_0_, and the shape of these isotherms corresponds to type I according to the IUPAC nomenclature [53], which is typical of microporous objects.

Modification of PVC with GO or RGO additives changes the form of isotherms, which includes a number of features typically associated with micro/mesoporous materials. Therefore, at very low relative pressures P/P_0_ < 0.1, the adsorption isotherm exhibited a steep rise associated with micropore filling, but the presence of a hysteresis loop indicates capillary condensation in mesopores. For the PCCNC modified with the addition of GO, the hysteresis loop was observed in the region above P/P_0_ ≥ 0.85 and a strong increase in the adsorbed amount close to saturation pressure resulted from pore condensation into large meso- and macropores. For PCCNC with the RGO additive, the hysteresis loop of the adsorption–desorption isotherm corresponds to type H4, where a lower limit of the desorption branch is normally located at the cavitation-induced P/P_0_ = 0.45, and this sharp step-down of the desorption branch at P/P_0_ < 0.4–0.5 is typical of nitrogen at 77 K. This type of isotherm can often be observed on micro/mesoporous carbons.

The analysis of the presented isotherms made it possible to obtain the main textural characteristics for both the PCCNC containing NC (NGC, GO or RGO) and the unmodified CM (Table 1). 

It should be noted that S_BET_ calculated from the selected linear range on the BET plot for solids containing micropores cannot be considered as the true probe accessible surface area, but rather the “apparent surface area” that is characteristic of the carbon materials under study [52,53]. According to the data presented above, the use of PVC in the CM synthesis without modifying additives allows, predominantly, the microporous material (the fraction of mesopores does not exceed 15%) with a specific surface area of 1102 m^2^ g^−1^ to be obtained. The presence of NGC in PCCNC does not produce substantial changes in the pore space structure of the material (the content ratio of micro- and mesopores remains virtually constant) (Figure 8) but somewhat (within 20–30%) decreases all parameters of the pore structure—both the S_BET_ and the specific pore volume. This may be caused by the contribution of NGC particles, having their own specific surface area of ca. 200 m^2^ g^−1^, to the composite properties. However, a much greater (~300 m^2^ g^−1^) decrease in the composite surface area, as compared to that derived from the additive model (~50 m^2^ g^−1^), may indicate that carbonization of polyvinylene molecules near the carbon nanoglobule surface leads to the formation of a carbon matrix with a higher density than in the absence of NC. This may be evidenced by different forms of the radial contrast intensity profiles (Figure 3d and Figure 4f), a comparison of which (Appendix A) reveals a smaller contribution of 1.4–0.8 nm distances to the RCIP of the composite. Hence, this PCCNC contains a smaller amount of strongly amorphized carbon than unmodified CM, thus decreasing both the composite surface area and the micropore volume.

The introduction of GO or RGO, as was noted above, results in an additional development of mesoporous component of the pore space during the synthesis of PCCNC. Thus, the pore size distribution curves for CM with GO or RGO additives, which were obtained with application of the BJH method to the adsorption branch (Figure 8), have a broad mesopore size distribution in the region of 10–200 nm and maxima on the distribution curve at 100 and 150 nm, respectively, for CM with RGO or GO additives. Thus, the formation of additional pore space in the presence of graphite oxide additives occurs mostly due to large meso- and macropores; this is more typical of PCCNC with the GO additive.

The formation of materials with a greater fraction of mesopores in the case of GO or RGO additives is confirmed by TEM data—the observed voids in the amorphous carbon matrix and “crumpled sheet” structures may testify to the developed mesoporous structure (Figure 5b and Figure 6b).

Evidently, the observed radical differences in the pore structure of PCCNC are associated with differences in the properties of modifying additives. As was noted above, the employed graphite oxide additives, in distinction to NGC, undergo thermal transformations in the temperature region used in our study for carbonization and activation of the samples, which are related to decomposition of the oxygen-containing groups in these materials and a release of gaseous products. According to data on thermal decomposition products of GO and RGO in a vacuum [60], upon heating to 400 °C (carbonization), GO evolves ca. 3.5 mmol g^−1^ CO_2_ + H_2_O as well as ca. 1.5 mmol g^−1^ CO, whereas RGO (the synthesis of which includes a short-term action) evolves a smaller amount of gases: 0.47 mmol g^−1^ CO_2_ + H_2_O, and virtually no CO. A further elevation of temperature to 900 °C can potentially lead to an additional release of 1.5 mmol g^−1^ CO_2_ and 3.5 mmol g^−1^ CO in the case of GO, and 0.15 mmol g^−1^ CO_2_ and 3.5 mmol g^−1^ CO in the case of RGO. The observed (Table 1) sharp increase in the fraction of mesopores in the presence of GO and RGO additives (up to 46% for GO and 36% for RGO) at close values of specific surface area and specific volume of micropores is caused by the release of gaseous products from graphite oxides, which facilitates the formation of a more loose structure of the carbon matrix. Therewith, the produced aggregates of graphene structures, which have a higher ordering of graphene layers in comparison with the carbon matrix, may stabilize (reinforce) the highly porous structure of the final carbon–carbon nanocomposite.

The specific volume and structure of micropores (the distribution of micropores by their size and morphology) play a significant role in the formation of the properties of adsorbents; therefore, in this work, attention was paid to the study of the microporous component of the obtained CM using the NLDFT method. The analysis of the micropore size distribution NLDFT curves allows us to conclude that the form of the curves is similar for CM without modifying additives and all types of the obtained PCCNC. The micropores are nonuniform in size: regions of ultramicropores (d = 0.5–0.7 nm) and supermicropores (d = 0.7–2.0 nm) were observed on the distribution curves (Figure 9). This result suggests that carbon obtained from PVC, which is present in all the samples under consideration, is responsible for the formation of micropores.

## 4. Conclusions

Micro- and mesoporous carbon–carbon nanocomposites were synthesized by alkaline dehydrochlorination of polyvinyl chloride in a dimethyl sulfoxide medium in the presence of graphite oxide, reduced graphite oxide or nanoglobular carbon dispersions with subsequent thermal treatment (carbonization, carbon dioxide activation) of the formed carbon-enriched structures. The main emphasis in the work is on the electron microscopic characterization of solids and the study of their porous structure.

The TEM study has revealed that during the synthesis of porous carbon–carbon nanocomposites (PCCNC), the introduced nanostructured components (NC) retain their original structure: the “crumpled sheet” aggregates in the case of graphite oxide additives or carbon nanoglobules in the case of carbon black additive, but affect the formation of the microporous carbon matrix of PCCNC arising from PVC. Mainly, such an influence is reflected in the change in the porous structure of PCCNC (the development of meso- and macroporosity). It can be assumed that this is due to changes in the contributions of pore formation routes: micro- and mesopores are formed during “burnout” under the action of CO_2_ of the most disordered carbon (mainly when NC is NGC), whereas macropores can be formed due to “internal gas release” during decomposition of GO.

According to the low-temperature nitrogen adsorption data, the synthesized materials are characterized by the specific surface area S_BET_ from 791 to 1115 m^2^ g^−1^ and the fraction of mesopores from 11% to 46% depending on the features of the introduced nanostructured component. The results obtained demonstrate feasibility of the controllable synthesis of porous carbon materials with specified textural characteristics using a certain modifier that is selected within the proposed approach to the formation of carbon structures of nanocomposite type.

It should be noted that the ratio of pores of different sizes will depend not only on the nature but also on the amount of the additive. At the same time, additivity in changing properties should not be expected, since synergistic effects and reaching the percalation threshold are possible. Graphene layers formed from PVC are short and strongly curved in different directions. This structure of the particles of the synthesized carbon material is due to the morphological features of the precursors of the carbon material—aggregates of entangled PVC molecules. During the formation of the composite, there is probably a mutual influence of both components on each other. Thus, a more ordered carbon structure is formed from PVC at the boundaries of contact with the surface of a more structured additive, and at the same time, the carbon matrix formed from PVC limits the region of GO and ROG transformation, reducing the length of their graphene layers. These processes, occurring with a change in volumes, lead to the formation of defects and the beginning of the formation of a porous space. Further development of porosity occurs at the stage of high-temperature gasification. Undoubtedly, these questions are fundamental and are the subject of further research.

The synthesis of PCCNC using graphite oxide and reduced graphite oxide as the modifying additives can be considered as a method for synthesizing a porous carbon material with the hierarchical structure containing both the micro- and meso/macropores. Such materials are widely used to create adsorbents and catalysts (they allow improving mass transfer and accessibility of reagents and overcoming steric and diffusion limitations in catalytic reactions) and have demonstrated their great potential for the application in energy storage systems. In addition, the reported results can be useful in the development of a dioxin-free process for the utilization of PVC and its wastes in order to obtain marketable micro- and mesoporous carbon materials with controllable structural properties and wide application sphere. In accordance with this, the next stages are to increase the efficiency and environmental friendliness of the synthesis of carbon composites, as well as to study their specific properties that are important for solving applied objectives.

## Figures and Tables

**Figure 1 materials-15-07636-f001:**
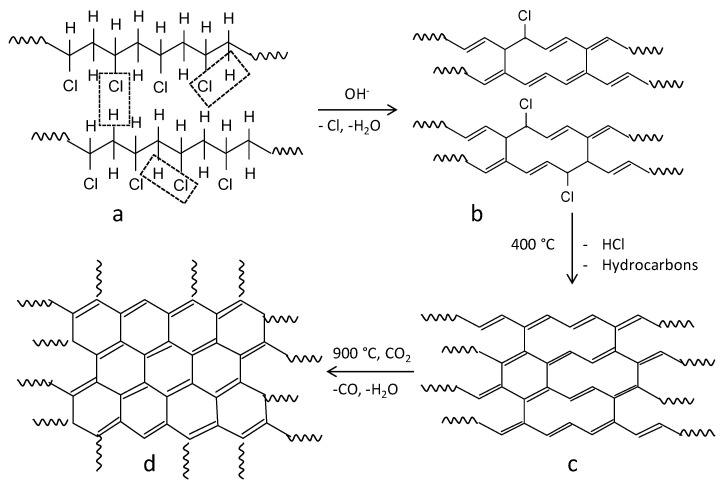
A scheme of CM synthesis via alkaline dehydrochlorination of PVC, low-temperature carbonization and activation (cited from [58] and rearranged): (**a**) a fragment of the initial PVC molecule; (**b**) fragments of polyvinylene chains formed upon hydrodechlorination of PVC molecules; (**c**) a fragment of the condensation product of polyvinylene chains, (**d**) a carbon material formed by the high-temperature treatment in CO_2_ medium.

**Figure 2 materials-15-07636-f002:**
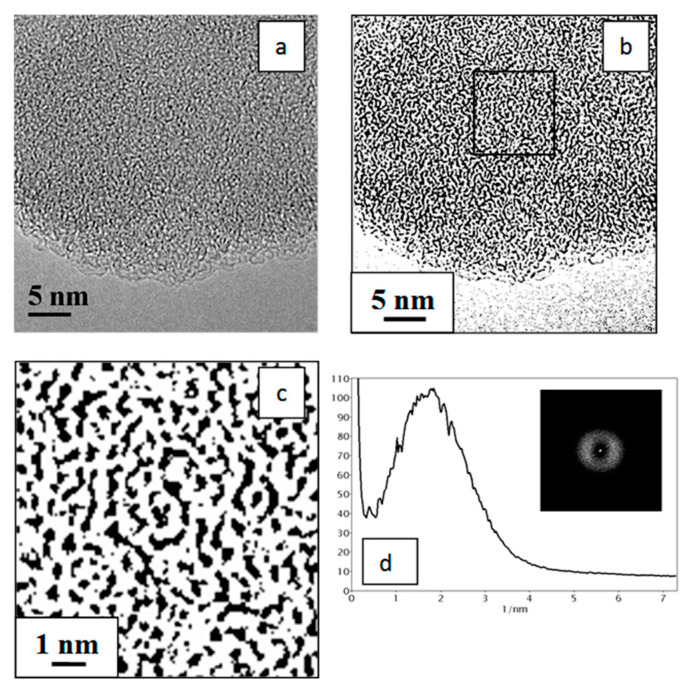
TEM image of the carbon material obtained by thermal treatment of dehydrochlorinated PVC at 400 °C (**a**); the same TEM image in the binary form (**b**); its scaled-up central part marked by a square (**c**); and the radial contrast intensity profile (RCIP) (**d**), obtained by processing the FFT image that is displayed in the inset (units on the x axis correspond to the inverse distance between graphene layers, 1/nm).

**Figure 3 materials-15-07636-f003:**
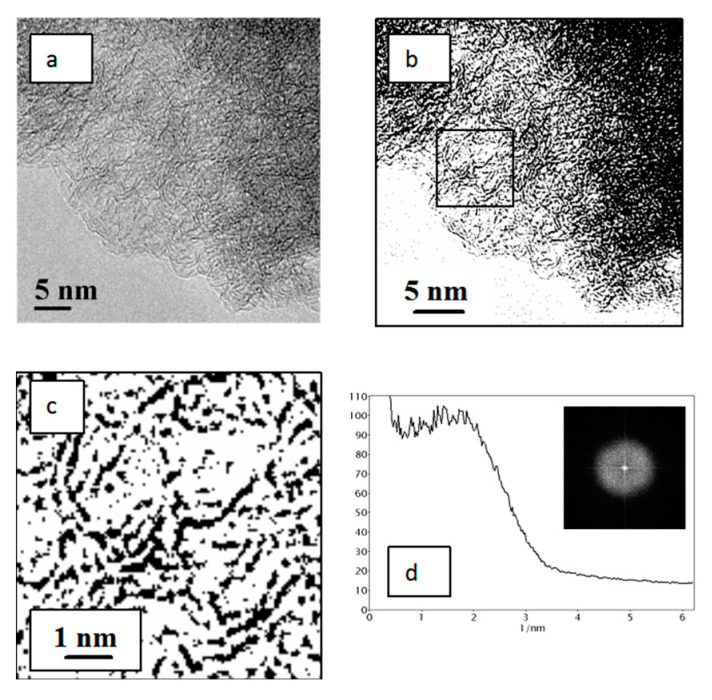
TEM image of the carbon material after thermal treatment at 900 °C (CO_2_) of the CM sample obtained by carbonization at 400 °C of dehydrochlorinated PVC (**a**); the same TEM image in the binary form (**b**); its scaled-up central part marked by a square (**c**); RCIP (**d**) obtained by processing the FFT image that is displayed in the inset (units on the x axis correspond to the inverse distance between graphene layers, 1/nm).

**Figure 4 materials-15-07636-f004:**
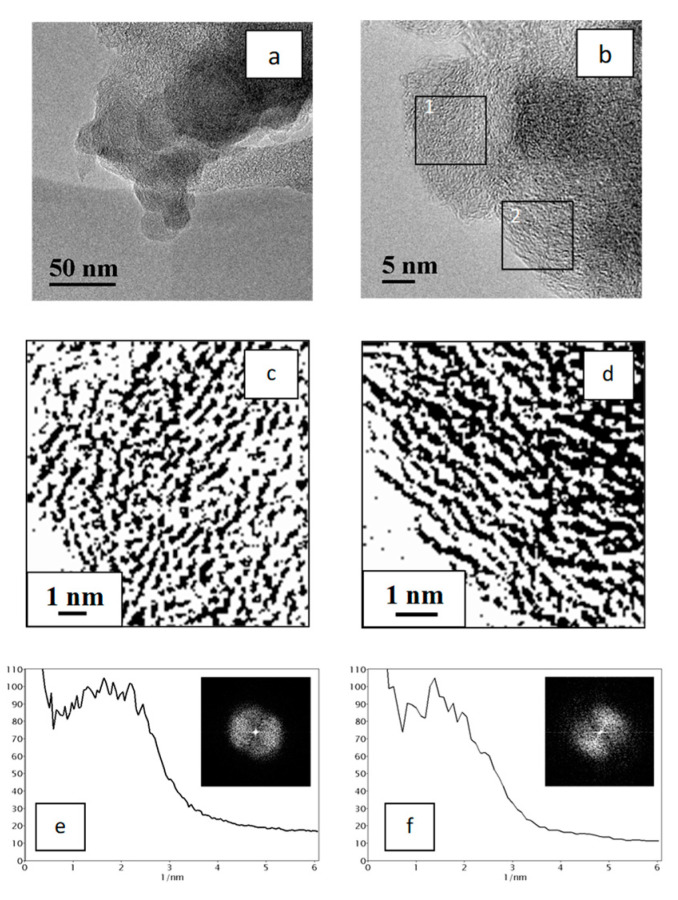
TEM image of PCCNC with the aggregate of NGC globules residing in the carbon matrix (**a**); TEM image with the NGC region (fragment 1) and the carbon matrix region (fragment 2) marked by squares (**b**); fragment 1 (**c**) and fragment 2 (**d**) in the binary form; RCIP obtained by processing the FFT images displayed in the insets: fragment 1 (**e**) and fragment 2 (**f**) (units on the x axis correspond to the inverse distance between graphene layers, 1/nm).

**Figure 5 materials-15-07636-f005:**
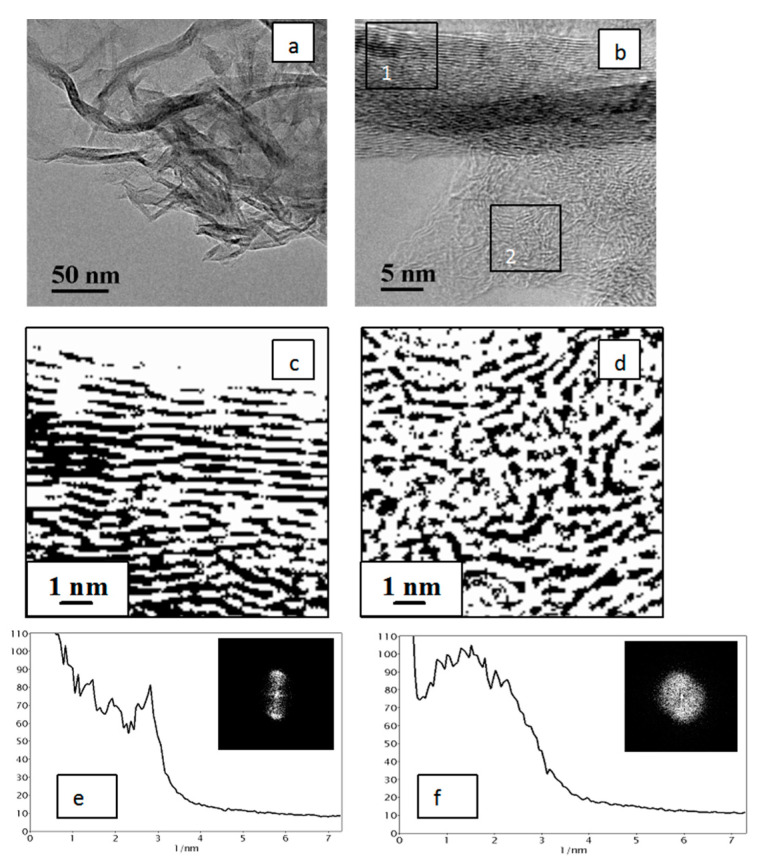
TEM images of PCCNC obtained using the GO additive (**a**,**b**); fragments of the TEM image (**b**) in the binary form (marked by squares): fragment 1 (**c**) and fragment 2 (**d**); RCIP obtained by processing the FFT images displayed in the insets (**e**,**f**) (units on the x axis correspond to the inverse distance between graphene layers, 1/nm).

**Figure 6 materials-15-07636-f006:**
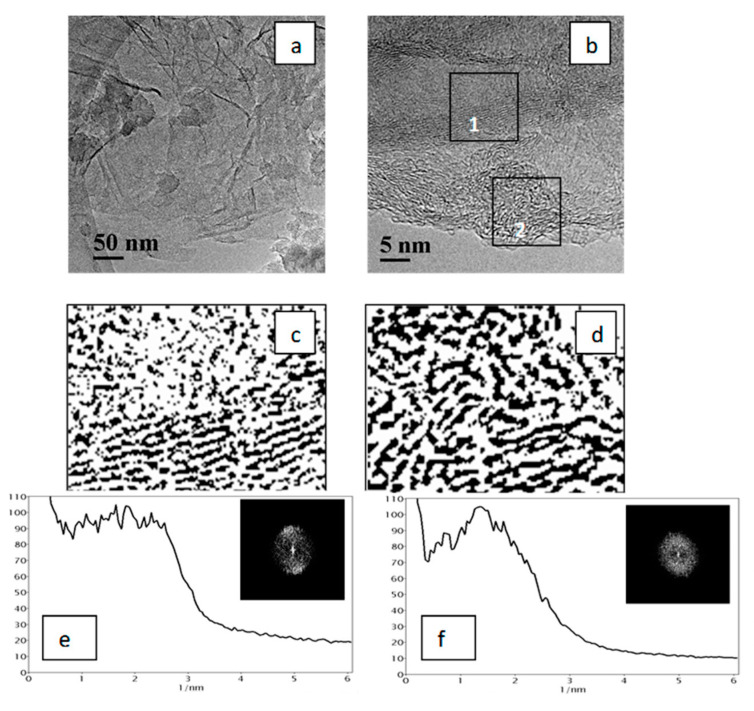
TEM images of the PCCNC obtained using the RGO additive (**a**,**b**); fragments of the TEM image (**b**) in the binary form (marked by squares): fragment 1 (**c**) and fragment 2 (**d**); RCIP obtained by processing the FFT images displayed in the insets (**e**,**f**) (units on the *x* axis correspond to the inverse distance between graphene layers, 1/nm).

**Figure 7 materials-15-07636-f007:**
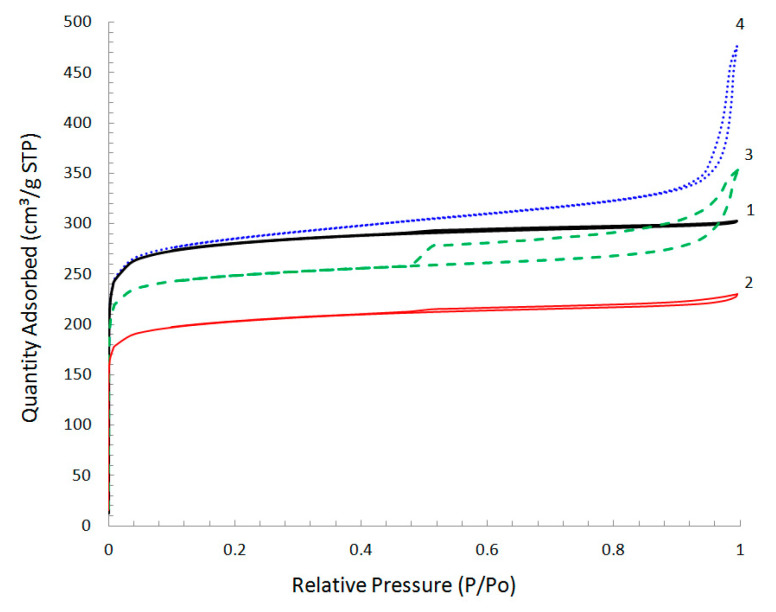
Low-temperature nitrogen adsorption–desorption isotherms for CM without additives (1); PCCNC with the NGC additive (2); PCCNC with the RGO additive (3); PCCNC with the GO additive (4).

**Figure 8 materials-15-07636-f008:**
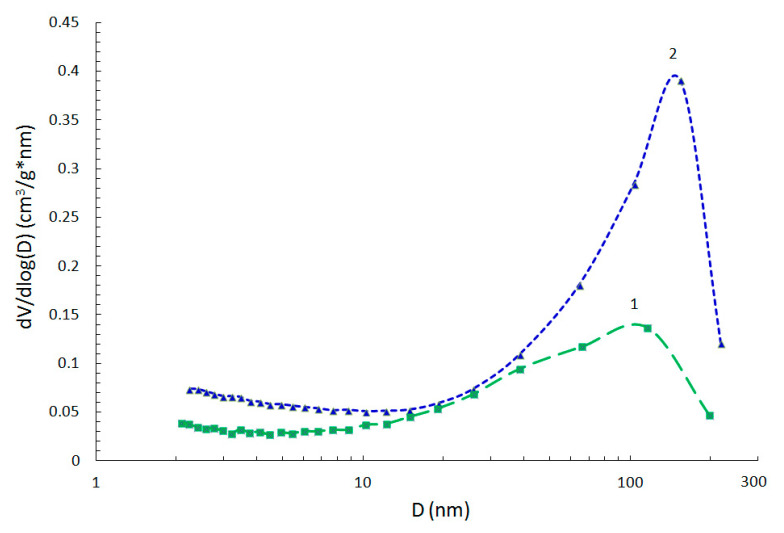
Pore size distribution curves for PCCNC with RGO (1) or GO (2) additives.

**Figure 9 materials-15-07636-f009:**
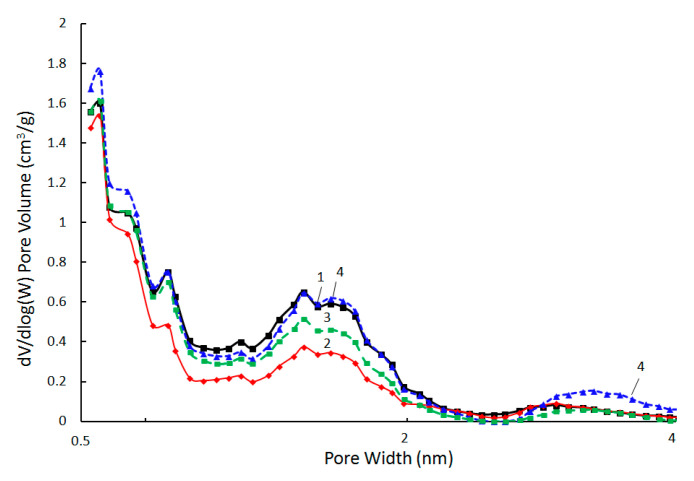
The micropore size distribution curves obtained within NLDFT method using the model for slit-like pores (CM without additives (1) and PCCNC with NGC (2), RGO (3) or GO (4) additives).

**Table 1 materials-15-07636-t001:** Pore structure parameters of the carbon material and PCCNC based on PVC.

Modifying Additive (NC Content, wt.% ^1^)	S_BET_, m^2^ g^−1^	V_ads_, cm^3^ g^−1^	V_micro_, cm^3^ g^−1^		Fraction of Meso (Macro) Pores, rel. %
			t-Method	NLDFT	
-	1102	0.47	0.40	0.42	15
NGC (6)	791	0.35	0.31	0.30	11
GO (5)	1115	0.70	0.38	0.41	46
RGO (7)	979	0.55	0.35	0.37	36

^1^ The content of modifying carbon additive in the composite after carbonization and activation steps.

## Data Availability

Data are contained in the article and Appendix A. Any additional data are available on request from the corresponding author.

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
