# Peer review of "Porous Carbon–Carbon Composite Materials Obtained by Alkaline Dehydrochlorination of Polyvinyl Chloride"

_materials, 2022, doi:10.3390/ma15217636_

Round 1

Reviewer 1 Report

In this manuscript, the authors reported an interesting way to prepare carbon composite materials. It is funny. The experiment process is also very detailed.  It can be accepted after a minor revision.

1. According to the results, can I infer that I can get composite with different aperture parameters by changing the amount of addition GO, for example, I want to get material with more or less microporous materials.  2. The authors said its morphology had a disordered pattern of graphene layers. Was it possible the morphology was resulted from the addition of GO or RGO rather than from PVC. 3. It may be better if the conclusion could be more general, and the highlights of the article may be highlighted more directly in the introduction. 4. Pay attention to sentence segmentation in the full text, such as "The TEM study has revealed that during the synthesis of porous......" 5. The format of references must be consistent. For example, article 6 and 7. There is something wrong with the article 18, please check it.

Author Response

Our general response:

The authors thank the Reviewer for a careful analysis of the presented material. We checked and edited the text of the manuscript and tried to take into account all the recommendations.

  1. According to the results, can I infer that I can get composite with different aperture parameters by changing the amount of addition GO, for example, I want to get material with more or less microporous materials.

Thank you for raising this issue. You are absolutely right that the ratio between pores of different sizes will depend on both the nature and the amount of the additive. However, one should not expect additivity in changing properties, since synergistic effects and reaching the percalation threshold are possible. Undoubtedly, this issue is fundamental and requires further detailed study. We have added information about the importance and necessity of such a study to the Conclusion.

  1. The authors said its morphology had a disordered pattern of graphene layers. Was it possible the morphology was resulted from the addition of GO or RGO rather than from PVC.

Graphene layers formed from PVC are short and strongly bent in different directions. Such a structure of particles of the synthesized carbon material is due to the morphological features of the precursors of the carbon material - aggregates of tangled PVC molecules. During the formation of the composite, there is probably a mutual influence of both components on each other. Thus, a more ordered structure of carbon is formed from PVC at the boundaries of contact with the surface of a more structured additive, and at the same time, the carbon matrix formed from PVC limits the region of transformation of GO and ROG, reducing the length of their graphene layers.

  1. It may be better if the conclusion could be more general, and the highlights of the article may be highlighted more directly in the introduction.

Thanks for the recommendation. In accordance with them, we have made changes to the Conclusion

  1. Pay attention to sentence segmentation in the full text, such as "The TEM study has revealed that during the synthesis of porous......"

Thank you for your advice. We have divided and simplified long sentences where possible.

  1. The format of references must be consistent. For example, article 6 and 7. There is something wrong with the article 18, please check it.

Thank you, we checked the references and made the necessary changes.

Reviewer 2 Report

In this work, the authours have investigated carbon-carbon nanocomposite synthesis by alkaline dehydrochlorination of PVC in the presence of GO, rGO or other carbon dispersions at high temperature by thermolysis.

From TEM, BET, and FFT data, their morphlogies were extensively investigated. However, more convincing data, e.g. thermal, electrical, dielectrical, optical properteis should be presented to cover the possible applications or improved characteristics of carbon-carbon composites.

Reviewer 3 Report

This manuscript is dedicated to a fabrication scheme for carbon-carbon composite materials synthesized by the alkaline dehydrochlorination of polyvinyl chloride solutions. Not only the proposed fabrication scheme is an interesting and original approach to both practical and reliable synthesis of C-based inherently-nanostructured-containing composite materials, which have not been studied well yet. The structural characterization efforts (by TEM) are very well systematized and convincing. The discussion provided is quite adequate for the present purpose. The well detailed and at the same time comparative context of the results clarifies convincingly the researched synthesis scheme and prompts a good understanding how to synthesize porous carbon materials with hierarchical structure which can benefit wide range of developing research and new applications.

From practical point of view, the reported results thus bring new knowledge and certainly represent an original contribution in the present context.

The authors chose an adequate structure of the manuscript – an excellent point of departure for such a study. Also, they provided a balanced realistic and nicely illustrated presentation of their results and corresponding analysis that is of much scientific and practical interest and adds new knowledge to the field.

The present manuscript is a significant contribution, this work once published would be instructive and suggestive in terms of further studies and to a wider readership.

There are some minor issues with this already excellent manuscript that will need to be addressed before becoming suitable for publication, i.e., it can be considered for publication after a minor revision:

1: Title can be improved to better reflect the message of the paper; Porous (carbon) should be mentioned in the title.

2: Abstract should also include that main characterization effort to which most conclusions are based is TEM.

3: In the introduction, the authors miss that a wide range of nanostructured and/or complex C-based materials have been subject to theoretical studies that assist understanding synergies and synthesis issues, including preparation schemes, and even directly guide corresponding experimental work. Examples, based on synthetic growth concept include ACS applied materials & interfaces 10 (2018) 16238-16243 and Vacuum 182 (2020) 109775. Such works should be referenced.

4: Authors should mention, elaborate more and be more specific about the inter-relation between bonding (e.g., C-C bonds), defects/disordered defect packings (as in graphene) and porosity.  

5: Spell-check and stylistic revision of the paper are still necessary. Some, long sentences, misspellings, etc., still are noticeable throughout the text.

Author Response

Reviewer 2

This manuscript is dedicated to a fabrication scheme for carbon-carbon composite materials synthesized by the alkaline dehydrochlorination of polyvinyl chloride solutions. Not only the proposed fabrication scheme is an interesting and original approach to both practical and reliable synthesis of C-based inherently-nanostructured-containing composite materials, which have not been studied well yet. The structural characterization efforts (by TEM) are very well systematized and convincing. The discussion provided is quite adequate for the present purpose. The well detailed and at the same time comparative context of the results clarifies convincingly the researched synthesis scheme and prompts a good understanding how to synthesize porous carbon materials with hierarchical structure which can benefit wide range of developing research and new applications.

From practical point of view, the reported results thus bring new knowledge and certainly represent an original contribution in the present context.

The authors chose an adequate structure of the manuscript – an excellent point of departure for such a study. Also, they provided a balanced realistic and nicely illustrated presentation of their results and corresponding analysis that is of much scientific and practical interest and adds new knowledge to the field.

The present manuscript is a significant contribution, this work once published would be instructive and suggestive in terms of further studies and to a wider readership.

There are some minor issues with this already excellent manuscript that will need to be addressed before becoming suitable for publication, i.e., it can be considered for publication after a minor revision:

Our general response:

We are grateful to the reviewer for his/her positive overall evaluation of our article. We tried to take into account all the comments and, in accordance with them, made changes into the manuscript.

1: Title can be improved to better reflect the message of the paper; Porous (carbon) should be mentioned in the title.

The title of the article has been changed

2: Abstract should also include that main characterization effort to which most conclusions are based is TEM.

Thank you for your advice. This information has been added to the Abstract. “The focus of the study was on the analysis and digital processing of transmission electron microscopy images to study local areas of carbon composite materials, as well as to determine the distances between graphene layers”.

3: In the introduction, the authors miss that a wide range of nanostructured and/or complex C-based materials have been subject to theoretical studies that assist understanding synergies and synthesis issues, including preparation schemes, and even directly guide corresponding experimental work. Examples, based on synthetic growth concept include ACS applied materials & interfaces 10 (2018) 16238-16243 and Vacuum 182 (2020) 109775. Such works should be referenced.

Recommendations taken into account, these articles added to the list of References. We have also added several modern review articles related to the synthesis and properties of carbon-containing composite nanomaterials (Refs. 27, 28)

4: Authors should mention, elaborate more and be more specific about the inter-relation between bonding (e.g., C-C bonds), defects/disordered defect packings (as in graphene) and porosity.

In the process of obtaining a composite, its components are subjected to interaction at the interface, as a result, the character of the interaction between carbon atoms changes in this region. A more ordered structure is formed from less ordered (amorphous) carbon (in the case of PVC), and a less ordered structure is formed from more ordered structures (in the case of GO and RGO, the carbon matrix formed from PVC limits the region of transformation of GO and ROG, reducing the length of their graphene layers).These processes, which occur with a change in volumes, lead to the formation of defects and the beginning of the formation of a porous space. Further development of the porous space occurs at the stage of high-temperature gasification.

5: Spell-check and stylistic revision of the paper are still necessary. Some, long sentences, misspellings, etc., still are noticeable throughout the text.

A highly qualified specialist checked all the text; the necessary changes and corrections were made.

Round 2

Reviewer 2 Report

Although, I still have doubts, the authors have answered most of the concerns. I recommend this to be published.